# A Prenatal DHA Test to Help Identify Women at Increased Risk for Early Preterm Birth: A Proposal

**DOI:** 10.3390/nu10121933

**Published:** 2018-12-06

**Authors:** Kristina H. Jackson, William S. Harris

**Affiliations:** OmegaQuant, LLC, 5009 W. 12th St., Suite 8, Sioux Falls, SD 57106, USA; bill@omegaquant.com

**Keywords:** DHA, omega-3 fatty acids, pregnancy, preterm birth

## Abstract

Fish intake and docosahexaenoic acid (DHA), a nutrient found in fish, have been favorably linked to several pregnancy outcomes. The risk of early preterm birth (ePT, <34 weeks gestation) is associated with low fish intake and DHA blood levels and can be reduced by supplemental DHA. Here, we summarize the evidence linking blood DHA levels with risk for ePT birth, and based on the available studies, propose that women who are pregnant or trying to become pregnant aim for a red blood cell (RBC) DHA value of at least 5% (of total RBC fatty acids). In the US, ~70% of women of childbearing age are likely below this cut-point, and dietary intake data suggest that this group, including pregnant women, consumes ~60 mg/day DHA and that >90% of this group do not take an omega-3 supplement. Since the recommendations for women to consume fish and to take a 200 mg DHA supplement during pregnancy are not being heeded generally, there is a need to motivate practitioners and pregnant women to attend to these recommendations. Having an objective prenatal blood DHA test could provide such motivation. More research is needed to test the clinical utility of this proposed target prenatal DHA level.

## 1. Introduction

Preterm (PT, <37 weeks gestation) birth is now the second leading cause of death in children under five worldwide and accounts for half of all newborn deaths, according to a report by the March of Dimes and the World Health Organization (WHO) in 2012 [1]. Early PT (ePT, <34 weeks gestation) birth puts infants at even higher risk for death and increases their risk for a variety of adverse health outcomes that can lead to extended time in the neonatal intensive care unit [2] and higher hospital costs [3]. PT births have increased in the last 20 years in almost all countries [4], and while treatment for PT infants has improved significantly, preventing PT birth and carrying a fetus to term is still safer, healthier, and cheaper. Known risk factors for PT birth are generally few and mostly non-specific. According to the WHO, these include history of PT birth, underweight, obesity, diabetes, hypertension, smoking, infection, younger or older age (i.e., under 17 or over 40), multi-fetal pregnancy (twins and higher), and pregnancies spaced too close together [1]. While there is clearly a need to address these risk factors, there may be at least one nutritional deficiency involved as well: a low intake of the long-chain omega-3 fatty acid, docosahexaenoic acid (DHA).

A prominent researcher in the area of preterm birth has recently written, “In my opinion, the most important conclusion that comes out from this study is the need for plasma eicosapentaenoic acid (EPA) + DHA concentration monitoring during pregnancy” [5]. What prompted this strong statement? A new study from Denmark demonstrated a 10-fold increase in risk for ePT birth in women with low omega-3 fatty acid blood levels [6]. Since known risk factors are so non-specific, options for prevention are relatively limited. The potential for a blood-based biomarker to provide some patient-specific information on risk for ePT birth is exciting, but not unexpected. The relationship between fish per se and/or a nutrient found in fish, DHA, and length of gestation has been studied for over 30 years. The evidence for the role of DHA in delaying spontaneous delivery has grown stronger in the last 10 years, leading now to the question of application—how do we use this nutritional information to lower risk for ePT birth?

The Academy of Nutrition and Dietetics recommends ~500 mg/day DHA in the prenatal diet, via low-mercury fish intake (2 servings per week) but does not make an explicit recommendation for supplementation [7]. The March of Dimes [8], the Food and Agriculture Organization of the UN [9], the International Society for the Study of Fatty Acids and Lipids [10], and the World Association of Perinatal Medicine [11] all recommend at least 200 mg/day DHA from either fish or supplements. However, this advice does not appear to be reaching the pregnant population, where the habitual intake of DHA appeared to be only ~60 mg/day and only 9% of pregnant women in the US report taking an omega-3 supplement [12]. Notably, the PT birth rate in the US in 2010 was higher than similar Western countries (per 100 live births, 12% vs. 7.8% Canada, 7.8% UK) [1]. In this paper we endeavor to lay the scientific foundation for establishing a preliminary prenatal blood DHA target that can be measured during pregnancy and would identify women at increased risk of ePT birth. Women thus identified could be especially encouraged by their healthcare providers to increase their DHA intake, whether from fish or by supplementation.

## 2. Relationship between Maternal DHA Levels and Preterm Birth

### 2.1. Epidemiology

The topic of PT birth and omega-3 fatty acids has been the focus of a research team led by Dr. Sjúrdur Olsen in Denmark at the Centre for Fetal Programming in the Statens Serum Institut for over 30 years, including epidemiological studies and large randomized controlled trials. Their most recent study was a nested case-control study measuring early and mid-pregnancy plasma levels (percent composition) of omega-3 fatty acids eicosapentaenoic acid (EPA) + DHA in women who had ePT births vs. term births [6]. Women in the lowest (first) quintile of plasma EPA + DHA had a 10-fold increased risk of ePT birth compared to the top 3 quintiles (adjusted Odds Ratio [OR] 10.27, 95% confidence interval [CI]: 6.80, 15.79). Those in the second quintile had ~3-fold increase in risk for ePT birth (adjusted OR 2.86, 95% CI: 1.79, 4.59). Using the average of the first and second trimester plasma EPA + DHA levels (which differed by about 7%) and converting them to red blood cell (RBC) DHA levels based on data from our laboratory resulted in the following values for the latter metric: Quintile 1 (Q1): 1.39–3.49%; Q2: 3.49–3.88%; Q3–5: 3.88–5.98%. There appeared to be a threshold effect at ~4–5% RBC DHA (converted from 2–2.5% plasma EPA + DHA) where the risk of ePT birth was reduced relative to lower levels, but additional protection was not provided with higher blood levels (Figure 1). To our knowledge, this is the only epidemiological study that has studied blood levels of fatty acids and risk of ePT birth, and it suggests that an RBC DHA of >4–5% might be a clinically useful metric.

In epidemiological studies that only use dietary intake data, higher fish intake and/or omega-3 supplementation has been associated with lower PT or ePT birth rates in some but not all studies. For example, Brantsaeter et al. found that Norwegian women reporting 1–2 servings of fish per week had 24% (Hazard Ratio [HR] 0.76, 95% CI: 0.66, 0.88) lower risk of PT birth compared to those reporting no intake [13]. There was little added benefit to higher intakes, and the pattern was stronger for lean fish versus fatty fish. Reported omega-3 supplement use was also associated with lower rates of ePT birth (<32 weeks; *p* = 0.05). A pooling study of 19 European birth cohorts found that women who consumed fish at least once per week had lower rates of PT birth than those who ate fish rarely (<1 time per week) (HR 0.87, 95% CI: 0.82, 0.92), again with no extra benefit at higher intakes [14]. Olsen and Secher found that odds of PT birth were increased by a factor of 3.6 (95% CI: 1.2, 11.2) from the lowest fish intake group (never) to the highest (one time per week) in 8729 pregnant women in Denmark [15]. This was not confirmed, however, in a US cohort of over 2000 women, where estimated g/day EPA + DHA from dietary questionnaires was not related to PT birth or gestation length [16]. Other than providing a signal that higher fish intakes are generally associated with lower risk for PT birth, these diet-record-based studies give little information regarding what a target prenatal RBC DHA level might be.

### 2.2. Randomized Controlled Trials: Low-Risk Populations

The main randomized controlled trial (RCT) that has reported both prenatal RBC DHA levels and ePT birth rates is Carlson et al. 2013 [17]. This study tested whether 600 mg/day DHA would increase maternal DHA blood levels during pregnancy (and in cord blood), and affect gestation duration, birth weight and length as compared to a placebo group supplemented from <20 weeks gestation to birth. The average RBC DHA at the end of the study in the placebo group was 4.7 ± 1.3% and 7.3 ± 2.2% in the treatment group. Compared with the control group, the treated group experienced fewer ePT births (placebo: 4.8% vs. DHA: 0.6% of births; *p* = 0.03), fewer very low birthweight infants (<1500 g; placebo: 3.4% vs. DHA: 0% of births; *p* = 0.03), and fewer days spent in the hospital (placebo: 40.8 ± 44.0 days vs. DHA: 8.9 ± 10.1 days, *p* = 0.03). However, there was no effect on rates of PT births (placebo: 8.8% vs. DHA: 7.8% of births, *p* = not significant [NS]). In addition, the primary endpoints were also linked to RBC DHA levels during pregnancy, with higher levels in the mother and cord blood being associated with longer gestation duration and greater birth weights, lengths, and head circumferences (*p*-values < 0.05). This study would argue for a target RBC DHA level of at least >5%.

While there is a dearth of data on prenatal blood DHA levels and risk for ePT birth, there have been several large RCTs giving fish oil or DHA supplements to pregnant women for a variety of outcomes, which have been compiled in meta-analyses. Makrides and colleagues recently updated a Cochrane review on this topic, including 70 RCTs with over 19,000 participants [18]. The authors reported a significant 42% reduction in ePT birth risk (Relative Risk [RR] 0.58, 95% CI: 0.44, 0.77) based on findings in nine RCTs (*n* = 5204) and a 11% reduction in PT birth risk (RR 0.89, 95% CI: 0.81, 0.97) from 26 RCTs (*n* = 10,304). A probable increased risk for prolonged gestation (>42 weeks; RR 1.61, 95% CI: 1.11, 2.33) from six RCTs (*n* = 5141) was noted in the fish oil or DHA supplemented groups. They did not find any significant effects of fish oil or DHA on post-term induction rates, maternal adverse events, postpartum depression, or child development and cognition. A 2016 meta-analysis including over 10,000 pregnancies found that the fish oil group had a significant 22% reduction in risk of ePT birth (RR 0.78, 95% CI: 0.64, 0.95) and marginally significant 10% reduction in risk of PT birth (RR 0.90, 95% CI: 0.81, 1.00), as well as ~6 days longer gestation [19]. Another 2015 meta-analysis found risk for ePT birth was reduced by 58% (RR 0.42, 95% CI: 0.27, 0.66) and PT birth by 17% (RR 0.83, 95% CI: 0.70, 0.98) by the fish oil intervention, regardless of dose, timing of supplementation, or pregnancy risk status [20]. A 2012 meta-analysis on this same topic found that women taking a fish oil or DHA supplement during pregnancy had a 26% reduced risk of ePT birth (RR 0.74, 95% CI: 0.58, 0.94) and a trend toward decreased PT birth risk (RR 0.91, 95% CI: 0.82, 1.01) [21]. The authors also found that supplementation increased birthweight by 42.2 g (95% CI: 14.8, 69.7) and had a lower risk of low birthweight (RR 0.92, 95% CI: 0.83, 1.02). There appears to be a consistent effect of fish oil or DHA supplementation on duration of gestation, and on ePT birth specifically.

One of the seminal RCTs in this field is the DOMInO (DHA to Optimize Mother Infant Outcome) study conducted in Australia by Makrides and Gibson and colleagues [22]. This study included 2399 pregnant women and tested the effects of 800 mg/day of DHA during pregnancy (<21 weeks gestation to birth) on maternal postpartum depression and child cognitive and language development. Unfortunately, only cord—not maternal—blood DHA levels were reported, and the latter (earlier in pregnancy) cannot be reliably inferred from the former. Compared to the control group, the DHA group had a lower rate of ePT birth (placebo: 2.25% vs. DHA: 1.09% of births, *p* = 0.03), fewer admissions to the neonatal intensive care unit (placebo: 3.08% vs. DHA: 1.75% of births, *p* = 0.04), fewer low birthweight infants (<2500 g; placebo: 3.41% vs. DHA: 5.27% of births, *p* = 0.003), and higher birthweights (*p* = 0.03). However, there were also more post-term inductions or pre-labor cesarean deliveries in the DHA group (placebo: 13.72% vs. DHA: 17.59%, *p* = 0.01), indicating prolonged gestation. PT birth tended to be lower in the DHA group, but this difference was not significant (placebo: 7.34% vs. DHA: 5.60% of births, *p* = 0.09). The primary endpoints of this study (postpartum depression and childhood language development) were not affected by increased DHA intake during pregnancy.

### 2.3. Randomized Controlled Trials: High Risk Pregnancies

A 2007 meta-analysis of high-risk pregnancies found that fish oil treated groups had a lower rate of ePT birth than the controls (RR 0.39, 95% CI: 0.18 0.84); however, because the rate of having a very low birth weight (<2500 g) infant was not decreased commensurate with the lower ePT birth rate, they recommended caution in interpretation [23]. There were no effects on PT birth rates in this population.

Two large RCTs in women with high-risk pregnancies tested whether supplemental fish oil is safe and if it can improve a variety of unfavorable pregnancy outcomes. A multi-center study [24] from nine sites in Europe included pregnant women with a variety of conditions, grouped into two main cohorts: prophylactic and therapeutic. The prophylactic cohort included 232 women with previous PT birth, 280 with previous intrauterine growth retardation (IUGR), 386 with previous pregnancy-induced hypertension, and 579 with twins in the current pregnancy. The therapeutic cohort included 79 with pre-eclampsia and 63 with suspected IUGR in the current pregnancy. Women in the prophylactic cohort were treated with 2.7 g/day fish oil (~900 mg/day DHA) from 20 weeks gestation to birth, and in the therapeutic cohort with 6.1 g/day fish oil (~2100 mg/day DHA) from 33 weeks gestation to birth. A placebo (olive oil) group was included in each cohort. The prophylactic group (except the twins cohort) receiving fish oil had a reduction in PT rates from 21–33% (OR 0.54, 95% CI: 0.30, 0.98) and reduced occurrence of ePT birth (OR 0.32, 95% CI: 0.11, 0.89). In addition, the fish oil group had a significantly longer gestation (8.5 days, 95% CI: 1.9, 15.2), greater occurrence of post-term birth (>42 weeks; OR 1.27, 95% CI: 1.08, 1.49), and higher birthweights (*p* = 0.02). There were no effects of fish oil on pregnancy duration in the twins cohort. In the therapeutic cohorts, the fish oil group had a delay in spontaneous delivery (HR 2.0, 95% CI: 1.16, 3.45), which was also true for the entire study population (HR 1.22, 95% CI: 1.07, 1.39, *p* = 0.0002). Finally, in a reanalysis of the previous study, Olsen and colleagues found that baseline fish intake interacted with the effects of fish oil on PT birth in that those with the low or medium reported intake who were in the treatment group had a 44% (HR 0.56, 95% CI: 0.36, 0.86) and 39% (HR 0.61, 95% CI: 0.44, 0.84) lower risk of PT birth compared to those in the placebo group [25]. There was no significant effect in those with high fish intake at baseline, further supporting the theory that having higher levels of DHA in the body, regardless if it is from fish or supplements, affects the process of spontaneous delivery in singleton births. Unfortunately for our present purpose, no blood levels of DHA were reported.

A similar US-based study by Harper et al. also tested the effects of fish oil on PT birth rates in a high-risk population [26]. Pregnant women with previous PT birth (*n* = 852) were randomized to either fish oil or a placebo from <22 weeks to 36 weeks gestation. The dose of fish oil the treatment group received was 2 g/day (~800 mg/day DHA), similar to the dose from Olsen et al. 2000 [24]. Although trends were observed, there was no significant difference in PT birth (RR 0.91, 95% CI: 0.77, 1.07) or ePT birth (<35 weeks; RR 0.95, 95%CI: 0.72, 1.25) between the groups; however, all women in the study were also treated with a drug known to delay PT birth, 17alpha-hydroxyprogesterone caproate, as a standard of care, which complicates interpretation. Plasma levels of EPA, DHA, and arachidonic acid were measured in 512 women at enrollment and mid-pregnancy. A significant increase in DHA plasma levels occurred in the treatment group (est. 1.6% increase; *p* < 0.0001) with a slight decrease in the placebo group (est. 0.3% decrease); however, unfortunately, only a graph of the change in percent composition was presented with no actual fatty acid values reported, so estimated RBC DHA levels cannot be derived.

In a secondary analysis of the Harper study, Klebanoff et al. stratified risk for PT birth by reported fish intake and by RBC DHA at baseline in 701 women [27]. Rates of ePT birth were not included in this analysis. The average baseline RBC DHA was 3.55% and 3.73% in the treatment and placebo groups, respectively. Women in the lowest quartile of RBC DHA at baseline (Q1: <3.05% vs. Q2–4: 3.05–>4.43%) had a significantly higher rate of PT birth (Q1: 47.2% vs. Q2–Q4: 34.9–43.4%, *p* = 0.03) compared to the three other quartiles, which was attenuated somewhat while controlling for supplementation status and other covariates (OR 1.41, 95% CI: 0.97, 2.05). Similarly, the women with the lowest reported fish intake (<1 serving/month) had a higher risk of PT birth than those who ate fish more often (48.6% vs. 35.9%, *p* < 0.001), which was still significant after controlling for the same variables. Consuming three fish meals per week was associated with the lowest odds for PT birth (OR 0.60, 95% CI: 0.38, 0.95), with no further benefits seen at higher intakes (although there were much fewer women at higher intakes). Also, there was no difference in the effect of the fish oil on PT birth based on baseline RBC DHA levels or fish intake.

### 2.4. Upcoming Trials

Two large RCTs are currently underway testing the effect of 800–1000 mg/day DHA on ePT birth, one in Australia and one in the US. The first, called the ORIP Trial [28] (Omega-3 fats to Reduce Incidence of Prematurity) recruited 5544 women from across Australia in early pregnancy (<20 weeks gestation) and randomized them to either an 800 mg/day DHA supplement or a placebo until 34 weeks gestation. The primary outcome is ePT birth, and maternal RBC DHA levels are reportedly being measured at recruitment and 34 weeks. The study completed recruitment in December 2017, therefore, results are forthcoming. The second study is called the ADORE Study [29] (Assessment of DHA on Reducing Early preterm birth). It is recruiting 1200 women in the US in early pregnancy (<20 weeks gestation) and is also testing the effect of DHA on ePT birth, but the dose in the active group is 1000 mg/day and in the control group, 200 mg/day (the latter is the commonly recommended DHA dose in pregnancy). Maternal RBC DHA will be measured, and the study should be complete by 2021. The results of these two studies could significantly change prenatal DHA recommendations, at least in Western countries, and provide greater insight regarding target prenatal RBC DHA levels.

### 2.5. Other Maternal and Child Outcomes Related to DHA

Although we have focused on DHA and ePT birth, there is evidence that higher (vs. lower) DHA levels/intakes during pregnancy are associated with a variety of favorable health outcomes. For example, DHA or fish oil supplementation has had beneficial effects in the following areas: in term and PT infants—visual and neurodevelopment [17,30,31], sleep quality [32], autonomic function [33]; and in toddlers and children—cognitive function [34,35,36], ability to focus [37], risk of asthma and wheezing [38], and overall growth [39]; and, in the mothers—reduced risk of postpartum depression [22,40] and higher DHA levels in breastmilk [41].

## 3. Is an RBC DHA of >5% a Reasonable Target?

At present, there are insufficient data to firmly establish a target prenatal RBC (or blood or plasma) DHA level. However, based primarily on the two studies reporting both blood levels and ePT rates [6,17], there is reasonable support for setting a preliminary, lower threshold for RBC DHA at 5% to help reduce risk for ePT birth. This is consistent with the characterization by Carlson et al. [29] of RBC DHA levels in pregnant women of 4.3% as “very low” and 3.5% as being “exceedingly deficient.”

This means if a pregnant woman (or one trying to conceive) has an RBC DHA of <5%, she should be encouraged (or prescribed) to increase her DHA intake, either by consuming 8–12 oz (225–340 g) per week of low-mercury, high-DHA fish [7] and/or taking a prenatal DHA supplement, preferably at least 600–800 mg/day DHA. A case could also be made for an even lower threshold of 3%, since that level has been linked to an even higher risk of ePT birth (Q1 from Olsen et al. was <3.5% [6]), and such levels are not uncommon in high risk pregnancies (e.g., Q1 from Harper et al. was <3% [26,27]). For these women, the recommended DHA intake should probably be >1000 mg/day DHA [24]. If the RBC DHA is >5%, then she should be encouraged to continue with her current dietary and supplementation habits, including meeting the current recommendations of at least 200 mg/day DHA, throughout the rest of the pregnancy. Having higher RBC DHA levels (e.g., 6.5–8%, the equivalent of 8–12% for the Omega-3 Index (RBC EPA + DHA)), appears to be safe and likely desirable. Indeed, mean RBC DHA levels from studies in non-supplemented pregnant women have been: 7.3% in Japan [42], 6.0% and 8.3% in Norway [35,43], and 6.7% in Australia [44].

Two studies in pregnancy, albeit with different outcomes, help confirm the efficacy of a 5% threshold RBC DHA during pregnancy. First, Markhus et al. [40] found that pregnant Norwegian women in the lowest 25% percentile of RBC EPA + DHA levels (5.1%; RBC DHA est. 4.6%) had a higher risk of postpartum depression than those above that threshold. Second, Bisgaard et al. [38] found that pregnant Danish women in the lowest third of whole blood EPA + DHA levels (<4.3%; RBC DHA est. 4.5%) benefited from an EPA + DHA supplement, in that their children were at reduced risk for asthma at 6 years old compared to placebo, while those with higher baseline levels did not benefit from the supplementation. These studies illustrate a potential “DHA deficiency state” during pregnancy that can manifest in a variety of outcomes for mother, fetus, and child.

In the US, the average RBC DHA level in women between the ages of 20 and 40 is about 3.7% based on data from 10,815 individuals [45]. Although the number of pregnant women included in this dataset is unknown, this level is likely to be fairly representative of both pregnant and non-pregnant women because the intake of fish in pregnant and non-pregnant women in this age group is similar [12]. DHA supplementation rates are different between pregnant and non-pregnant women in the US but are both low at 9% vs. 2%, respectively [12]. Therefore, we estimate that about 70% of the US women of childbearing age are below the proposed threshold RBC DHA value of 5%.

## 4. Are There Risks Associated with an RBC DHA >5% in Pregnancy?

As noted above and to the best of our knowledge, RBC DHA levels up to ~9% are not associated with increased risk for any adverse outcomes for mother or infant [35]. Indeed, supplementation studies using up to 6 g of fish oil (>2000 mg DHA) have not reported any increased risk for serious complications, including excessive bleeding, during pregnancy or delivery [24,35].

Although it is clear that having higher vs. lower RBC DHA levels is good for pregnant women, determining exactly how high and how to safely achieve such levels is somewhat challenging. Dietary supplements containing DHA (with or without EPA, e.g., fish oils) can give some women “fishy burps” but there are no health risks associated with them. On the other hand, choosing the right fish to consume is more complicated. Because a few species of fish contain levels of methyl mercury that are potentially dangerous for the developing fetus, pregnant women have been instructed to not only avoid raw fish (sushi, for food safety reasons), but high-mercury fish as well. As a consequence, many women simply forgo fish altogether during pregnancy [46]. Indeed, food aversions and nausea during pregnancy may also make consuming fish difficult for some women. This topic is outside the scope of this review, but Food and Drug Administration (FDA) and Environmental Protection Agency guidelines have changed recently to emphasize the benefits of fish intake during pregnancy while at the same time providing guidance on avoiding specific species high in mercury [47]. Finally, higher DHA intakes and/or blood levels have been associated with longer gestation or increased birthweight [17,30,31]. This can result in either post-term cesarean or inductions [18,25]. Although clearly not as serious as ePT birth, this is nonetheless undesirable. The ORIP Study is testing whether stopping DHA supplementation at 34 weeks gestation will help mitigate these prolonged gestation outcomes. Despite the potential negative consequences of prolonged pregnancy, this effect nevertheless supports the hypothesis that DHA plays a physiological role in gestation, the mechanisms of which (briefly discussed below) should be further explored.

## 5. How Might a Target RBC DHA Level Be Used in Obstetric Practice?

To our knowledge there are currently no other laboratory tests that predict risk of ePT birth, especially ones that can be done in an office or at home via dried blood spot sample collection. Whole blood or dried blood spot samples can be sent to a laboratory where they are treated to produce fatty acid methyl esters that are then analyzed by gas chromatography, as recently described [48]. Furthermore, an RBC DHA test is actionable, that is, it can readily be changed by consuming more DHA, which is both safe and, as reviewed above, likely to be beneficial to mom and baby even when taken only during the second part of pregnancy.

Ideally, women would be tested for RBC DHA levels and correct any deficiencies prior to becoming pregnant, however, this may not be realistic since 45% of all pregnancies in the US are unplanned [49]. Since prenatal care requires several clinic visits throughout the pregnancy, RBC DHA testing could easily be incorporated into the current prenatal care process. Based on current data, the testing could be done in the first or second trimesters [6], which affords sufficient time to make changes in DHA intake if needed. Many of the current studies have recruited subjects well into the second trimester and have had success with supplementation between mid-pregnancy and birth.

## 6. Why Not Just Recommend Higher DHA Intake to Everyone and Not Test?

There are already pregnancy recommendations for fish intake of 8–12 oz (225–340 g) per week [7] and a recommendation for 200 mg/day DHA from supplements [8], but as described above, these are apparently not being met. Based on the results from ORIP and ADORE, these recommendations may increase to 800 mg/day DHA or more, but the issue of motivation still remains. While these are important endorsements, the lingering effects of the FDA’s “fish toxicity” guidelines for pregnant women may have made women hesitant to eat fish during pregnancy [46]. However, if a simple blood test could show that DHA levels are low, and this can be linked to increased risk for ePT birth (as we have attempted to do here), then such information could provide the necessary motivation to consume more DHA, whether via fish or supplements. Furthermore, testing blood levels also allows practitioners to understand their patients’ risk and encourages healthy dietary practices that could have lasting effects on their children (i.e., lower risk of asthma [38]) and themselves (i.e., post-partum depression [40]).

It might be suggested to do a simple fish intake questionnaire or ask about supplement usage instead of testing, but we would caution that questionnaires cannot replace objective biomarker analysis. Somewhere between 20% and 50% of the variability in DHA blood levels can be explained with dietary intake questionnaires [50,51]. While blood DHA levels are associated with fish intake, there are several reasons why predicting them is difficult, including differences in the rates of absorption and tissue incorporation, types of fish eaten, misreporting/remembering, etc. Blood levels in pregnancy are also difficult to predict because fatty acids from adipose tissue stores, especially DHA, are released for the fetus and marginally change blood levels [52]. (Despite this natural process, DHA intake clearly affects blood DHA levels during pregnancy). Moreover, personalized recommendations to reach a target blood level of DHA will hopefully be more effective in motivating women to consume more DHA than blanket recommendations to eat more fish or take DHA supplements has been.

Prevention is not free, but it is much less costly than treatment. In this case, testing is relatively inexpensive, easy to do (via finger prick or blood draw), available (in the US), and the results are actionable. If DHA intake needs to increase, fish and supplements are inexpensive (as compared to a pharmaceutical or medical intervention) and already recommended by medical bodies. According to a cost–benefit analysis of the DOMInO study, the authors estimated that, if all pregnant women took DHA supplements, the Australian public hospital system could save between $15–51 million Australian dollars ($10–36 million US dollars) per year. A similar analysis was conducted by Shireman et al. [53] based on the study by Carlson et al. [17] in the US. These investigators calculated that giving 600 mg of DHA per day during the last two trimesters could save the healthcare system $6 billion US dollars per year. Thus, identifying women at especially high risk for ePT birth by RBC DHA testing and increasing compliance to DHA supplementation could have a significant health and economic impact. While more research is needed to better define the target level, the added motivation of seeing a blood level related to ePT birth risk may help increase the number of women meeting recommendations.

## 7. Possible Mechanisms for a DHA Effect on Early Preterm Birth

A thorough discussion of the potential biochemical basis for an effect of DHA on risk for ePT birth is beyond the scope of this paper. Briefly, as described by Norwitz et al. [54], uterine activation follows the increased expression of contraction-associated proteins (which include myometrial receptors for prostaglandins and oxytocin), activation of specific ion channels, and an increase in connexin 43 (a key component of gap junctions). An increase in gap junctions facilitates electrical synchrony within the myometrium and permits coordination of contractions. Once activated, the uterus can be stimulated to contract by the actions of oxytocin and the stimulatory prostaglandins E2 and F2α. DHA may slow this process at several steps via its effects on membrane physical properties [55], which may impact surface receptor activity. Omega-3 fatty acids also modulated the expression of connexin 43 in cardiac tissue of rats with experimental diabetes [56]. Perhaps most importantly, DHA (via its competition with arachidonic acid for conversion to a variety of oxylipins [57]) may reduce the levels of prostaglandins PGE2 and F2α.

## 8. Conclusions

Many lines of evidence now support the consumption of fish and DHA for a healthier pregnancy for both the mother and the infant. DHA and fish oil supplementation studies have shown that the omega-3 fatty acids do indeed have a biological effect on gestational duration and are safe. The current fish intake and DHA recommendations for pregnancy are not being heeded by a vast majority of pregnant women in the US. Personalized nutritional testing of prenatal DHA blood levels can provide needed guidance for pregnant women and their practitioners to identify the best supplemental dose of DHA for their situation or to create a plan to increase fish intake safely. A threshold level of 5% RBC DHA appears to be a reasonable initial target level to strive for, since risk of ePT birth seems to increase below this level. More research is needed (and is forthcoming) to test whether this threshold value is truly clinically relevant. Since increasing DHA intake (and RBC levels) in the second half of pregnancy can reduce said risk, consideration should be given to assessing RBC DHA levels during pregnancy and then acting on the results to optimize levels for the sake of both the mother and the baby.

## Figures and Tables

**Figure 1 nutrients-10-01933-f001:**
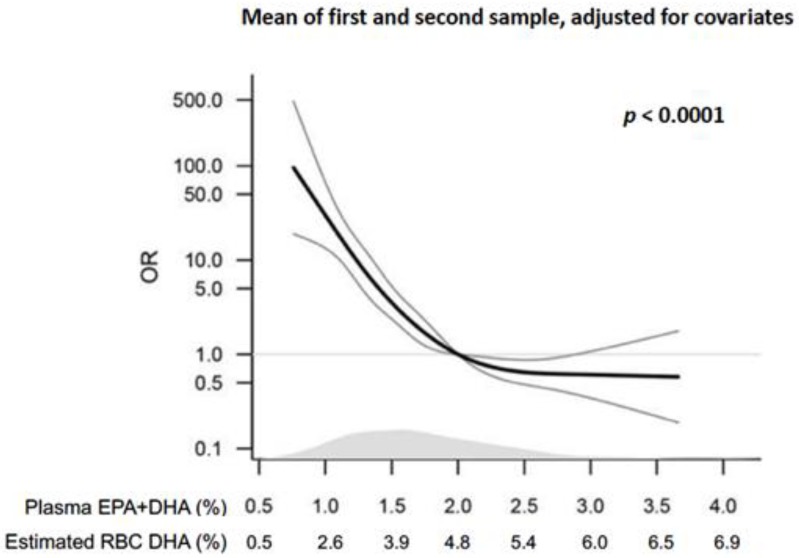
Odds ratios (OR) with 95% confidence intervals (CI) for early preterm birth given measurements of plasma EPA + DHA modelled as restricted cubic splines with five knots. Above the x-axis is shown the density of EPA + DHA measurements estimated in controls. Erythrocyte (RBC) DHA levels were estimated from plasma EPA + DHA (%) with the equation Y = 0.0306 × ln(x) + 0.1673. The equation is based on a sample of 2309 plasma and RBC samples analyzed at OmegaQuant (Sioux Falls, SD, US) in 2017–2018. Figure adapted from Olsen SF et al. 2018 EbioMedicine [6] and used with permission. EPA: eicosapentaenoic acid; DHA: docosahexaenoic acid; RBC: red blood cell.

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
