# Peer review of "A Prenatal DHA Test to Help Identify Women at Increased Risk for Early Preterm Birth: A Proposal"

_nutrients, 2018, doi:10.3390/nu10121933_

Reviewer 1 Report

The concept of having a set threshold level for RBC DHA is interesting and the papers you presented could potentially support a recommendation for measuring levels during pregnancy. However, be cautious about asserting that there needs to be recommended blood tests for DHA and an optimal RBC DHA level of 5%. The research presented in your paper does not support this recommendation. Please see below for further explanation and additional suggested edits:

 Include the biologic plausibility that higher levels of RBC DHA could result in less risk for preterm infants. While epidemiologic studies and RCTs have suggested there could be a link between DHA and improved birth outcomes, this is not sufficient proof in what you are suggesting. Especially since improved dietary intake and access to supplements is directly linked to higher SES, detailing the biologic rational for why increased n-3 PUFAs could affect pregnancy outcomes is important for your paper. More explanation is also needed on  whether RBC DHA levels fluctuate during pregnancy. I am concerned about this issue because you cited the Olsen study as the sole evidence behind the 5% RBC DHA threshold level. However this threshold is based on an average of first and second trimester which may invalidate the 5% threshold level you cited.

While your studies are extensive, visualizations would assist readers to contextualize your review of the literature. Include tables/forest plots documenting your studies and their findings. It would be particularly useful if you could present the exposure levels (ie, documented seafood consumption, amount of supplementation, trimester measured, etc.) in these tables and/or graphs.

 lines 58-65: the study is a nested case-control study (not a case-control study).

 Make sure you include CIs for the results that you report. For example, line 35 you presented a nonsignificant result, but I am curious how much overlap there was between the treatment and placebo groups.

 Lines 159-163: These sentences were confusing. Make sure you add your comparison group (ie, “those with the low or medium reported intake WHO WERE IN THE TREATMENT GROUP had a 44% (HR 0.56, 95% CI: 0.36, 0.86) and 39% (HR 0.61, 95% CI: 0.44, 0.84) lower risk of PT birth COMPARED TO THOSE IN THE PLACEBO GROUP”)

 You mention how women may forgo fish consumption during pregnancy because of mercury content. Also address the issue of nausea during pregnancy and how that could affect seafood consumption

 Author Response

Reviewer 1 (Responses in italics)

 Be cautious about asserting that there needs to be recommended blood tests for DHA and an optimal RBC DHA level of 5%. The research presented in your paper does not support this recommendation.

 We have tried to be cautious in laying out the rationale for setting SOME level, and even more so, in suggesting 5% as a preliminary target. In our view, the benefits to mom and baby of striving to achieve ANY RBC DHA level more than outweigh the risk associated with setting the “wrong” target. As regards the reviewer’s comment that “research presented in your paper does not support this recommendation” we would disagree. Perhaps s/he meant that it does not “fully support,” or “prove,” or “definitively confirm” such a recommendation; and we agree with that.

 Include the biologic plausibility that higher levels of RBC DHA could result in less risk for preterm infants.

 We agree that this was a gap in the present paper. We have now included a brief section on plausible mechanisms, the most reasonable of which is that DHA, via its cyclo-oxygenase, lipoxygenase, and CYP-450 oxylipin metabolites, counter the pro-parturition effects of the arachidonic acid derived oxylipins. See Section 7 (lines 341-353).  (To be clear, the evidence supports a role for higher maternal DHA in reducing risk for pre-term BIRTH; it’s not reduced risk for pre-term INFANTS).

 While epidemiologic studies and RCTs have suggested there could be a link between DHA and improved birth outcomes, this is not sufficient proof in what you are suggesting. Especially since improved dietary intake and access to supplements is directly linked to higher SES, detailing the biologic rational for why increased n-3 PUFAs could affect pregnancy outcomes is important for your paper.

 Whereas epidemiological evidence can indeed be confounded by SES and other factors, this is not true for RCTs, and there are RCT data (see section 2.3 of the manuscript) describing the RCT (and meta-analyses of the RCTs) showing that giving DHA (vs placebo) impacts gestation. True, RBC DHA is often not measured in these studies, but we have tried to show that with the doses given, levels in the neighborhood of 5% or higher were likely achieved. So this story does not rest entirely on epidemiologic data.

 More explanation is also needed on whether RBC DHA levels fluctuate during pregnancy. I am concerned about this issue because you cited the Olsen study as the sole evidence behind the 5% RBC DHA threshold level. However, this threshold is based on an average of first and second trimester which may invalidate the 5% threshold level you cited.

 In Olson, the difference between the first and second trimester plasma DHA levels averaged about 7% (not 7 percentage points). The main point is that estimating blood levels with fish or supplement intake questionnaires is less useful than measuring blood levels. We have now included a comment on this point in lines 68-69 of the revised manuscript.

 While your studies are extensive, visualizations would assist readers to contextualize your review of the literature. Include tables/forest plots documenting your studies and their findings. It would be particularly useful if you could present the exposure levels (i.e., documented seafood consumption, amount of supplementation, trimester measured, etc.) in these tables and/or graphs.

 We are including a figure adapted from Olsen et al. 2018 (permission requested, awaited response) that demonstrates exposure (DHA blood levels) and the primary outcome (early preterm birth OR).

 lines 58-65: the study is a nested case-control study (not a case-control study). Corrected.

 Make sure you include CIs for the results that you report. For example, line 35 you presented a nonsignificant result, but I am curious how much overlap there was between the treatment and placebo groups.

 We actually did include 95% CIs each time we reported a study’s outcome, whether a hazard ratio, and odds ratio, or an intervention result.  (We were unable to locate the result presented on line 35, so we cannot address this concern).

 Lines 159-163: These sentences were confusing. Make sure you add your comparison group (ie, “those with the low or medium reported intake WHO WERE IN THE TREATMENT GROUP had a 44% (HR 0.56, 95% CI: 0.36, 0.86) and 39% (HR 0.61, 95% CI: 0.44, 0.84) lower risk of PT birth COMPARED TO THOSE IN THE PLACEBO GROUP”) Corrected.

 You mention how women may forgo fish consumption during pregnancy because of mercury content. Also address the issue of nausea during pregnancy and how that could affect seafood consumption.

 We know of no data on nausea’s effect on the appetite for to seafood specifically. I assume that nausea should not be a problem post trimester 1 when the importance of DHA intake accelerates. 

Reviewer 2 Report

Brief summary

The manuscript highlights the importance of DHA on lowering the risk of preterm birth and the possibility of leveraging red blood cell DHA index on early deficiency detection. The manuscript is well-written with a clear tactic on review and discussion; at the end of the manuscript, the authors proposed and linked the threshold levels of the red blood cell DHA index with the amount of DHA that childbearing-age women may take to lower early preterm birth risk.

Specific comments

1. Introduction

Please briefly introduce what DHA index is and how it works; or, the authors can provide a reference here in case the readers would like to know more about this laboratory test and data interpretation.

What is the potential mechanism of action that DHA may lower the risk of preterm birth? Does EPA also contribute to lowering the risk or only DHA has the function? Please include relevant information in this section.

2. Relationship between maternal DHA levels and preterm birth

Line 207-215 [2.5 other maternal and child outcomes related to DHA]

Since this manuscript focuses on preterm birth, please remove unrelated content.

‍3. Is an RBC DHA of >5% a reasonable target?

Line 217 - the authors mentioned that "there are insufficient data to firmly establish a target RBC DHA level for pregnant women"; can the results/evidence from the ongoing ORIP trial and ADORE study help to strengthen their laboratory results with their diet/dietary supplement recommendations?

4. Are there risks associated with an RBC DHA >5% in pregnancy?

Line 264-265 - the authors mentioned that "the ORIP study is testing whether stopping DHA supplementation at 34 weeks gestation will help mitigate these prolonged gestation outcomes". How long a DHA-enriched diet and/or DHA dietary supplementation could alter the RBC DHA level? Also, is that possible that the pregnant women who stop DHA supplementation 5-6 weeks prior to delivery puts them in a high preterm risk again?

Author Response

Reviewer 2 (Responses in italics)

Introduction

Please briefly introduce what DHA index is and how it works; or, the authors can provide a reference here in case the readers would like to know more about this laboratory test and data interpretation.

 We have now included a brief discussion of the method and referred to the validation paper in lines 291-3.

 What is the potential mechanism of action that DHA may lower the risk of preterm birth?

 See earlier response to Reviewer 1 and Section 7 (lines 341-353).

 Does EPA also contribute to lowering the risk or only DHA has the function?

 We have no data from which to answer this question. Recent studies have used pure DHA and found an effect on ePT birth; no such studies have been conducted with pure EPA to the best of our knowledge.

 Line 207-215 [2.5 other maternal and child outcomes related to DHA].

Since this manuscript focuses on preterm birth, please remove unrelated content.

 Although this paper does focus on pre-term birth, part of the overall consideration regarding the pros and cons of setting a preliminary target RBC DHA levels must look at other possible risks and benefits to the dyad of doing so. Some of the potential benefits include the outcomes briefly discussed in this section, and we similarly commented on some of the possible down-sides (i.e., prolonged gestation past due date). We think it’s quite important to take as wide a look as we can at this question since, if RBC DHA targets were to be adopted, then the effects of such a move, whether for good or ill, have to be weighed.

 Line 217 - the authors mentioned that "there are insufficient data to firmly establish a target RBC DHA level for pregnant women"; can the results/evidence from the ongoing ORIP trial and ADORE study help to strengthen their laboratory results with their diet/dietary supplement recommendations?

 Absolutely! But these studies are not yet published, so until they are, their findings can’t inform this analysis.

 Line 264-265 - the authors mentioned that "the ORIP study is testing whether stopping DHA supplementation at 34 weeks gestation will help mitigate these prolonged gestation outcomes". How long a DHA-enriched diet and/or DHA dietary supplementation could alter the RBC DHA level?

 It normally takes about 3-4 months to reach a new steady state in RBC DHA after increasing DHA intake. Hence it makes sense to monitor a pregnant woman from the beginning of the 2nd trimester so that there would be sufficient time to make a meaningful change.

 Also, is that possible that the pregnant women who stop DHA supplementation 5-6 weeks prior to delivery puts them in a high preterm risk again?

 Unknown, but it seems unlikely since DHA has a long ½ life at least in non-pregnant persons.

Reviewer 3 Report

This is an interesting review on DHA supplementation and the rsk ot preterm delivery and I would like to congratulate with Authors for their effort

My criticism is the way of presentation of data and I would suggest to presenti in term of systematic review and meta analysis

Author Response

Reviewer 3 (Responses in italics)

My criticism is the way of presentation of data and I would suggest to presentation in term of systematic review and meta-analysis.

 Unfortunately, there are insufficient data sets to do a meta-analysis; hence we were forced to a simple review aimed at answering the question of whether a reasonable target might now be set for RBC DHA in pregnancy.

 Round  2

Reviewer 3 Report

I understand your point but I feel necessary that you report in the text your attempt (search in literature) to perform a systematic review and meta analysis

Author Response

We agree with the need for systematic reviews and meta-analyses in the review papers. However, here we are not attempting to perform a formal systematic review, but rather propose a concept. We tried where possible to report meta-analyses conducted by others, such as the recent Cochrane review on prenatal omega-3 supplementation and preterm birth (Middleton et al. 2018). However,  we did not find a meta-analysis on maternal blood fatty acid levels and preterm birth because blood levels have not been routinely measured in the supplementation trials or epidemiological studies. Therefore, how maternal blood levels of DHA affects early preterm birth rates is not easily found with typical search terms on PubMed. For example, we searched the terms "early preterm birth" and "maternal blood fatty acid levels" which resulted in 4 papers, none of which were useful for our question. We searched many of the DHA or fish oil supplementation studies in pregnancy for maternal blood fatty acid levels, but it was often not reported and whether or not blood levels were reported was typically not a searchable term. We hope in the future that maternal blood fatty acid levels will be consistently reported so that a systematic review can be performed.